# Normalizing Flows for Knockoff-free Controlled Feature Selection

**Derek Hansen**
Department of Statistics
University of Michigan
dereklh@umich.edu

**Brian Manzo**
Department of Statistics
University of Michigan
bmanzo@umich.edu

**Jeffrey Regier**
Department of Statistics
University of Michigan
regier@umich.edu

## Abstract

Controlled feature selection aims to discover the features a response depends on while limiting the false discovery rate (FDR) to a predefined level. Recently, multiple deep-learning-based methods have been proposed to perform controlled feature selection through the Model-X knockoff framework. We demonstrate, however, that these methods often fail to control the FDR for two reasons. First, these methods often learn inaccurate models of features. Second, the "swap" property, which is required for knockoffs to be valid, is often not well enforced. We propose a new procedure called FLOWSELECT to perform controlled feature selection that does not suffer from either of these two problems. To more accurately model the features, FLOWSELECT uses normalizing flows, the state-of-the-art method for density estimation. Instead of enforcing the "swap" property, FLOWSELECT uses a novel MCMC-based procedure to calculate p-values for each feature directly. Asymptotically, FLOWSELECT computes valid p-values. Empirically, FLOWSELECT consistently controls the FDR on both synthetic and semi-synthetic benchmarks, whereas competing knockoff-based approaches do not. FLOWSELECT also demonstrates greater power on these benchmarks. Additionally, FLOWSELECT correctly infers the genetic variants associated with specific soybean traits from GWAS data.

## 1 Introduction

Researchers in machine learning have made much progress in developing regression and classification models that can predict a response based on features. In many application areas, however, practitioners need to know *which* features drive variation in the response, and they need to do so in a way that limits the number of false discoveries. For example, in genome-wide association studies (GWAS), scientists must consider hundreds of thousands of genetic markers to identify variants associated with a particular trait or disease. The cost of false discoveries (i.e., selecting variants that are not associated with the disease) is high, as a costly follow-up experiment is often conducted for each selected variant. Another example where controlled feature selection matters is analyzing observational data about the effectiveness of educational interventions. In this case, researchers may want to select certain educational programs to implement on a larger scale and require confidence that their selection does not include unacceptably many ineffective programs. As a result, researchers are interested in methods that model the dependence structure of the data while providing an upper bound on the false discovery rate (FDR).

Model-X knockoffs (Candès et al., 2018) is a popular method for controlled variable selection, offering theoretical guarantees of FDR control and the flexibility to use arbitrary predictive models. However, even with knowledge of the underlying feature distribution, the Model-X knockoffs method is not feasible unless the feature distribution is either a finite mixture of Gaussians (Gimenez et al., 2019) or has a known Markov structure (Bates et al., 2020). Hence, a body of research explores the use of empirical approaches that use deep generative models to estimate the distribution of $X$ and

36th Conference on Neural Information Processing Systems (NeurIPS 2022).

sample knockoff features (Jordon et al., 2019; Liu & Zheng, 2018; Romano et al., 2020; Sudarshan et al., 2020).

The ability of these methods to control the FDR is contingent on their ability to correctly model the distribution of the features. By itself, learning a sufficiently expressive feature model can be challenging. However, the knockoff procedure requires learning a knockoff distribution that satisfies the *swap property*, which is a much stronger requirement. Formally, let $X \in \mathbb{R}^D$ be a sample from the feature distribution and $\tilde{X} \in \mathbb{R}^D$ be a sample from the knockoff distribution conditioned on $X$. The swap property stipulates that the joint distribution $(X, \tilde{X}) \in \mathbb{R}^{2D}$ must be invariant to swapping the positions of any subset of features $S \in \{1, \ldots, D\}$:

$$(X, \tilde{X})_{\text{swap}(S)} \overset{D}{=} (X, \tilde{X}) \tag{1}$$

Here, $\text{swap}(S)$ means exchanging the positions of $X_j$ and $\tilde{X}_j$ for all $j \in S$. For example, in the case $D = 3$ and $S = \{1, 3\}$, the joint distribution is $(X, \tilde{X}) = (X_1, X_2, X_3, \tilde{X}_1, \tilde{X}_2, \tilde{X}_3)$, and the swapped joint distribution is $(X, \tilde{X})_{\text{swap}(S)} = (\tilde{X}_1, X_2, \tilde{X}_3, X_1, \tilde{X}_2, X_3)$. Note that, for $S = \{1, \ldots, D\}$, the swap property implies that $\tilde{X} \overset{D}{=} X$. See Candès et al. (2018) for a more detailed description of the swap property.

Even if a distribution were found satisfying the swap property, it may not provide enough power to make discoveries. For example, both properties are trivially satisfied by constructing exact copies of the features as knockoffs, but the resulting procedure has no power.

In situations where a valid knockoff distribution is available to sample from, knockoffs are computationally appealing because they require only one sample from a knockoff distribution to assess the relevance of all $p$ features. However, in situations where the joint density of the features is unknown, we show that empirical approaches to knockoff generation (Jordon et al., 2019; Liu & Zheng, 2018; Romano et al., 2020; Sudarshan et al., 2020) fail to characterize a valid knockoff distribution and therefore do not control the FDR. We further show that even with a known covariate model, it is not straightforward to construct a valid knockoff distribution unless a specific model structure is known.

We propose a new feature selection method called FLOWSELECT (Section 3), which does not suffer from these problems. FLOWSELECT uses normalizing flows to learn the joint density of the covariates. Normalizing flows is a state-of-the-art method for density estimation; asymptotically, it can approximate any distribution arbitrarily well (Papamakarios et al., 2021; Kobyzev et al., 2020; Huang et al., 2018). Additionally, FLOWSELECT circumvents the need to sample a knockoff distribution by instead applying a fast variant of the conditional randomization test (CRT) introduced in Candès et al. (2018). Samples from the complete conditionals are drawn using MCMC, ensuring they are unbiased with respect to the learned data distribution.

Asymptotically, FLOWSELECT computes correct p-values to use for feature selection (Section 4). Our proof assumes the universal approximation property of normalizing flows and the convergence of MCMC samples to the Markov chain's stationary distribution. Under the same assumptions as the CRT, which includes a multiple-testing correction as in Benjamini & Hochberg (1995), a selection threshold can be picked which controls the FDR at a pre-defined level. Empirically, on both synthetic (Gaussian) data and semi-synthetic data (real predictors and a synthetic response), FLOWSELECT controls the FDR where other deep-learning-based knockoff methods do not. In cases in which competing methods do control the FDR, FLOWSELECT shows higher power (Section 5). Finally, in a challenging real-world problem with soybean genome-wide association study (GWAS) data, FLOWSELECT successfully harnesses normalizing flows for modeling discrete and sequential GWAS data, and for selecting genetic variants the traits depend on (Section 5.4).

## 2   Background

FLOWSELECT brings together four existing lines of research, which we briefly introduce below.

**Normalizing flows**   Normalizing flows is a general framework for density estimation of a multi-dimensional distribution with arbitrary dependencies (Papamakarios et al., 2021). A normalizing flow starts with a simple probability distribution (e.g., Gaussian or uniform), which is called the *base distribution* and denoted $Z$, and transforms samples from this base distribution through a

series of invertible and differentiable transformations, denoted $G$, to define the joint distribution of $X \in \mathbb{R}^D \sim \mathcal{P}_X$. A normalizing flow with enough transformations can approximate any multivariate density, subject to regularity conditions detailed by Kobyzev et al. (2020). Compared to other density-estimation methods, normalizing flows are computationally efficient. Details about the specific normalizing flow architecture used in FLOWSELECT are provided in Appendix A.

**Controlled feature selection**  Consider a response $Y$ which depends on a vector of features $X \in \mathbb{R}^D$. Depending on how the features are chosen, it is plausible that only a subset of the features contains all relevant information about $Y$. Specifically, conditioned on the relevant features in $X$, $Y$ is independent of the remaining features in $X$ (i.e. the null features). The goal of the controlled feature selection procedure is to maximize the number of relevant features selected while limiting the number of null features selected to a predefined level. If we denote the total number of selected features $R$, then we can decompose $R$ into $V$, the number of relevant features selected, and $S$, the number of null features selected.

**Conditional randomization test**  Controlled feature selection can be seen as a multiple hypothesis testing problem where there are $p$ null hypotheses, each of which says that feature $X_j$ is conditionally independent of the response $Y$ given all the other features $X_{-j}$. Explicitly, the test of the following hypothesis is conducted for each feature $j = \{1, \dots, D\}$:

$$H_0 : X_j \perp Y | X_{-j} \quad \text{versus} \quad H_1 : X_j \not\perp Y | X_{-j}. \tag{2}$$

To test these hypotheses, one can use a conditional randomization test (CRT) (Candès et al., 2018). For each feature tested in a conditional randomization test, a test statistic $T_j$ (e.g., the LASSO coefficient or another measure of feature importance) is first computed on the data. Then, the null distribution of $T_j$ is estimated by computing its value $\tilde{T}_j$ based on samples $\tilde{X}_j$ drawn from the conditional distribution of $X_j$ given $X_{-j}$. Finally, the p-value is calculated based on the empirical CDF of the null test statistics, and features whose p-values fall below the threshold set by the Benjamini-Hochberg procedure (Benjamini & Hochberg, 1995) are selected. Though the CRT is introduced as a computationally inefficient alternative to knockoffs, the CRT nonetheless has appeal because it requires only knowledge of the feature distribution, which can be learned empirically by maximum likelihood.

**Holdout randomization test**  The holdout randomization test (HRT) (Tansey et al., 2021) is a fast variant of the CRT; it uses a test statistic that requires fitting the model only once. Let $\theta$ represent the parameters of the chosen model, and let $T(X, Y, \theta)$ be an importance statistic calculated from the model with input data. For example, $T$, could be the predictive likelihood $\mathcal{P}_\theta(Y^{\text{test}}|X^{\text{test}})$ or the predictive score $R^2$. To use the HRT, first fit model parameters $\hat{\theta}$ based on the training data. Next, for each covariate $j$, calculate the test statistic $T_j^* \leftarrow T(X^{\text{test}}, Y^{\text{test}}, \hat{\theta})$. Then, generate $k$ null samples and compute $T_{j,k} \leftarrow T(X^{\text{test}}_{(j \leftarrow j_k)}, Y^{\text{test}}, \hat{\theta})$, where $X^{\text{test}}_{(j \leftarrow j_k)}$ replaces the $j$-th covariate with the $k$-th generated null sample. Finally, calculate the p-value as in the CRT, based on the empirical CDF of the null test statistics.

## 3  Methodology

FLOWSELECT implements the CRT for arbitrary feature distributions by using a normalizing flow to fit the feature distribution and Markov chain Monte Carlo (MCMC) to sample from each complete conditional distribution. Performing controlled feature selection with FLOWSELECT consists of the three steps below.

**Step 1: Model the predictors with a normalizing flow**

Starting with the observed samples of the features $X_1, \dots, X_N \sim \mathcal{P}_X$, we fit the parameters of a normalizing flow $G_\theta$ to maximize the log likelihood of the data with respect to a base distribution $p_Z$:

$$\hat{\theta} = \arg\max_{\theta} \sum_{i=1}^{N} \log p_{\theta}(X_i) \qquad (3)$$

$$\text{where } p_{\theta}(X_i) = p_Z(G_{\theta}(X)) \left| \det\left( \frac{\partial G_{\theta}(X)}{\partial X} \right) \right|.$$

The resulting density $p_{\hat{\theta}}$ is a fitted approximation to the true density $\mathcal{P}_X$. The specific normalizing flow architecture we use in our first two experiments consists of a single Gaussianization layer (Meng et al., 2020) followed by a masked autoregressive flow (MAF) (Papamakarios et al., 2017). The first layer can learn complex marginal distributions for each covariate, while the MAF learns the dependencies between them. More detail on normalizing flows and on this particular architecture can be found in Appendix A.

**Step 2: Sample from the complete conditionals with MCMC**

For each feature $j$, we aim to sample corresponding null features $\tilde{X}_{i,j,k}$ for all $k \in \{1, \ldots, K\}$ that are equal in distribution to $p_{\hat{\theta}}(X_{i,j}|X_{i,-j})$, but independent of $Y_i$. However, directly sampling from this conditional distribution is intractable. Instead, we implement an MCMC algorithm that admits it as a stationary distribution. The samples drawn from MCMC are autocorrelated, but any statistic calculated over these samples will converge almost surely to the correct value. The choice of the MCMC proposal distribution $q_j$ is flexible. Because each Markov chain is only one-dimensional, a Metropolis-Hastings Gaussian random walk with the standard deviation set based on the covariance can be expected to mix rapidly. Alternatively, information from $p_{\hat{\theta}}$, such as higher-order derivatives, could be used to construct a more efficient proposal. Algorithm 1 details how to implement step 2.

**Step 3: Test for significance with the HRT**

As in the CRT, feature $j$ has high evidence of being significant if, under the assumption that $j$ is a null feature, the probability of realizing a test statistic greater than the observed $T_j(X)$ is low. Formally, letting $[\tilde{X}_j, X_{-j}]$ be the observed feature matrix with the observed feature $X_j$ swapped out with the null feature $\tilde{X}_j$, we can write this as a p-value $\alpha_j$:

$$\alpha_j \equiv \mathcal{P}_{\tilde{X}_j|X_{-j}}\left( T_j(X) < T_j([\tilde{X}_j, X_{-j}]) \right). \qquad (4)$$

However, the above p-value $\alpha_j$ is not tractable. For each sample $\tilde{X}_{\cdot,j,k}$ drawn using MCMC, we calculate the corresponding feature statistic and compare it to the real feature statistic, leading to an

---

**Algorithm 1** Step 2 of the FLOWSELECT procedure for drawing $K$ null features $\tilde{X}_{i,j}|X_{i,-j}$ for feature $j$ at observation $i$.

---

**Input:** Feature matrix $X \in \mathbb{R}^{N \times D}$, observation index $i$, feature index $j$, number of samples $K$, fitted normalizing flow $p_{\hat{\theta}}$, MCMC proposal $q_j$
**Output:** Null features $\tilde{X}_{i,j,k}$ for $k = 1, \ldots, K$
**for** $k = 1, \ldots, K$ **do**
   Propose: $X_{i,j,k}^{\star} \sim q_j(\cdot | \tilde{X}_{i,j,k-1}, X_{i,-j})$
   $r_{i,j,k} \leftarrow \frac{p_{\hat{\theta}}(X_{i,j,k}^{\star}, X_{i,-j})q_j(\tilde{X}_{i,j,k-1}|X_{i,j,k}^{\star}, X_{i,-j})}{p_{\hat{\theta}}(\tilde{X}_{i,j,k-1}, X_{i,-j})q_j(X_{i,j,k}^{\star}|\tilde{X}_{i,j,k-1}, X_{i,-j})}$
   Sample: $U_{i,j,k} \sim \text{Bernoulli}(r_{i,j,k} \wedge 1)$
   **if** $U_{i,j,k} = 1$ **then**
      $\tilde{X}_{i,j,k} \leftarrow X_{i,j,k}^{\star}$
   **else**
      $\tilde{X}_{i,j,k} \leftarrow \tilde{X}_{i,j,k-1}$
   **end if**
**end for**

---

approximated p-value $\hat{\alpha}_j$:

$$\hat{\alpha}_j \equiv \frac{1}{K+1}(1 + \sum_{k=1}^{K} \mathbf{1}[T_j(X) < T_j([\tilde{X}_{j,k}, X_{-j}])). \tag{5}$$

To control the FDR, we use the Benjamini-Hochberg procedure to establish a threshold for the observed p-values. Specifically, we set the threshold to $s(\gamma) \triangleq \max_j\{\hat{\alpha}_j : \hat{\alpha}_j \leq \frac{j}{D}\gamma\}$, and select all features $j$ such that $\alpha_j \leq s(\gamma)$.

The Benjamini-Hochberg correction only guarantees FDR control provided that the p-values have either positive or zero correlation. Thus, the FDR control of FLOWSELECT depends on these assumptions being met. A more conservative correction from Benjamini & Yekutieli (2001) allows for arbitrary dependencies in p-values, but it suffers from low power. The Benjamini-Hochberg correction is widely used and empirically robust (Tansey et al., 2021), so we report results using it. Across our synthetic and semi-synthetic benchmarks in Section 5, we also find that FLOWSELECT maintains empirical FDR control.

Provided that the Benjamini-Hochberg assumptions are met, the FDR will be controlled, but the power of the test depends on $T_j$ being higher when $j$ is a significant feature. For example, if $Y$ is expected to vary approximately linearly with respect to $X$, $T_j(X)$ could be the absolute estimated regression coefficient $|\hat{\beta}_j|$ for the linear model $Y = X\beta + \epsilon$. Another choice is the HRT feature statistic described earlier.

## 4    Asymptotic results

The ability of FLOWSELECT to control the FDR relies on its ability to produce estimated p-values that converge to the correct p-values for the hypothesis test in Equation (2).

**Theorem 1.** *Let $X \in \mathbb{R}_{N \times D}$ be a random feature matrix, where each row $X_{i,\cdot}$ is independent and identically distributed; $x \in \mathbb{R}_{N \times D}$ be the observed feature matrix; and $\alpha_j$ be the p-value as defined in Equation (4) with test statistic $T_j(X)$. Suppose there exists a sequence of functions $(G^n)_{n=1}^{\infty}$ and a base random variable $Z$ satisfying the following conditions:*

  1. *Each $G^n$ is continuously differentiable and invertible.*

  2. *$G^n \to G$ pointwise for some map $G$ that is triangular, increasing, continuously differentiable, and satisfies $G(X_{i,\cdot}) \overset{D}{=} Z$.*

*For $n = 1, 2, \ldots$, let $X^n$ be the random feature matrix where each row $i$ is independent and has distribution $X_{i,\cdot}^n = (G^n)^{-1}(Z)$. Then, the p-value in Equation (5) calculated using $K$ MCMC samples targeting $X_{\cdot,j}^n \mid X_{\cdot,-j}^n = x_{\cdot,-j}$ converges to the correct p-value $\alpha_j$ with probability 1.*

Here we sketch the proof. A full proof can be found in Appendix B. First, by construction each $G^n$ defines a distribution $X_{i,\cdot}^n \overset{D}{=} (G^n)^{-1}(Z)$ that in turn implies a conditional distribution $X_{\cdot,j}^n | X_{\cdot,-j}^n = x_{\cdot,-j}$. We show these conditional distributions converge to the true conditional distribution of $X_{\cdot,j}$ given $X_{\cdot,-j} = x_{\cdot,-j}$. Consequently, the probability of observing a higher test statistic under the approximated null distribution $\tilde{X}_{\cdot,j}^n \overset{D}{=} X_{\cdot,j}^n$, written $\alpha_j^n$, will converge to the probability under the true null distribution $\tilde{X}_{\cdot,j}|X_{\cdot,-j} = x_{\cdot,-j}$, i.e. $\alpha_j$. Next, the Cesaro average of $K$ samples from an MCMC algorithm targeting $\tilde{X}_{\cdot,j}^n|X_{\cdot,-j} = x_{\cdot,-j}$, written $\hat{\alpha}_{j,K,n}$ will converge to $\alpha_j^n$ with probability 1 as $K \to \infty$. Combining these two convergences leads to the stated result.

Assuming the limiting p-values $\{\alpha_j\}$ satisfy the chosen multiple-hypothesis-testing assumptions, Theorem 1 specifies additional conditions that are sufficient for FDR control. These conditions are not strictly fewer than those required for empirical model-X knockoff-based methods to control FDR, but they may be easier to satisfy adequately in practice. For example, the condition that there exists a sequence $(G_n)_{n=1}^{\infty}$ converging to the true mapping $G$ is satisfied asymptotically by many flow architectures that are universal distribution approximators, including the Gaussianization Flows and Masked Autoregressive Flows used in our experiments (Huang et al., 2018; Meng et al., 2020; Kobyzev et al., 2020). In practice, it is unlikely that an exact mapping $G$ will be learned, as doing so

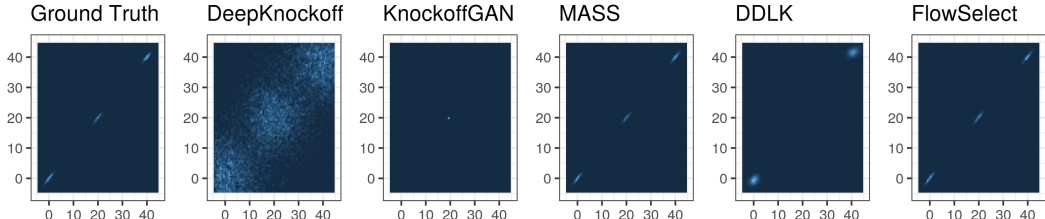

Figure 1: A density plot of the feature distribution with coordinate $j = 1$ on the x-axis and coordinate $j = 2$ on the y-axis. The ground truth density is compared to the normalizing flow fitted within FLOWSELECT and the distribution of each knockoff method (DeepKnockoff, KnockoffGAN, MASS, and DDLK). To have FDR control, each distribution should match the distribution of the features.

could require infinite training data, infinitely deep transformations, and exact nonconvex optimization. Nonetheless, normalizing flows work extremely well in practice; Theorem 1 gives intuition for the good performance of FLOWSELECT that we observe empirically.

## 5 Experiments

### 5.1 Synthetic experiment with a mixture of highly correlated Gaussians

We compare FLOWSELECT to the aforementioned knockoff methods with synthetic data drawn from a mixture of three highly correlated Gaussian distributions with dimension $D = 100$.[1] For each knockoff method, we use the exact implementation described in their respective papers, and we utilize the code made publicly available by the authors (c.f. Appendix D.3 for further details). For further comparison, we also implement the MASS knockoff procedure from Gimenez et al. (2019) and the RANK knockoff procedure from Fan et al. (2020). These methods estimate the unknown feature distribution using either a mixture of Gaussians (MASS) or a sparse precision matrix (RANK), and then sample the knockoffs directly as in Candès et al. (2018).

To generate the data, we draw $N = 100,000$ highly correlated samples. For $i = 1, \ldots, N$, we sample

$$X_i \overset{\text{i.i.d}}{\sim} \sum_{m=1}^{3} \pi_m p_{\mathcal{N}}(X_i; \mu_m, \Sigma_m), \tag{6}$$

with mixing weights $\pi = (0.371, 0.258, 0.371)$, mean vector $\mu = (0, 20, 40)$, and covariance matrices $\Sigma_m$. Each covariance $\Sigma_m$ follows an AR(1) pattern such that $(\Sigma_m)_{i,j} = \rho_m^{|i-j|}$ where $\rho = (0.982, 0.976, 0.970)$. The response $Y_i$ is linear in $f_i(X_i)$ for some function $f_i$ and coefficient vector $\beta$ i.e., $Y_i = f_i(X_i)\beta + \epsilon_i$. Each coefficient $\beta_j$ equals $\frac{100}{\sqrt{N}} B_j$, where $B_j = 0$ with probability 0.8, $B_j = 1$ with probability 0.1, and $B_j = -1$ with probability 0.1. We consider two different schemes for the $f_i$ that connect the features to the response. In our linear setting, $f_i$ is equal to the identity function. In our nonlinear setting, $f_i(x)$ is set equal to $\sin(5x)$ for odd $i$ and $f_i(x) = \cos(5x)$ for even $i$.

The experimental setting we have described so far is adapted from Sudarshan et al. (2020). However, we found that the $N = 2000$ they used was too few observations for any of the methods to do well in a general non-linear setting. Moreover, in many situations where controlled feature selection is deployed, neighboring features will be highly correlated. To reflect this, we also increased the base correlation between features within each mixture to create a more challenging example. We show results under the original settings of Sudarshan et al. (2020) in Appendix K.

For each model, we use 90% of the data for training to generate null features and the remaining 10% for calculating the feature statistics. To define the feature statistics, we use the holdout randomization test (HRT) described at the end of Section 2. For the HRT, we employ different predictive models for each response type ("linear" and "nonlinear"). Specifically, for the linear response, we use the predictive log-likelihood from the LASSO (Tibshirani, 1996), and for the nonlinear response, we use the predictive negative mean-squared error from a random forest regressor (Breiman, 2001).

---

[1]Software to reproduce our experiments is available at https://github.com/dereklhansen/flowselect.

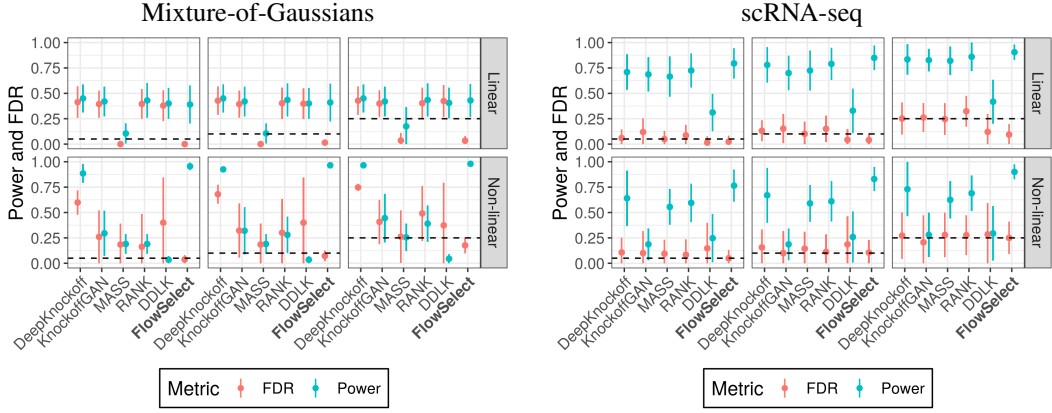

Figure 2: Comparison of power and false discovery rate (FDR) control of FLOWSELECT to knockoff methods on the Mixture-of-Gaussians dataset (left) and the scRNA-seq dataset (right) at targeted FDRs of 0.05, 0.1, and 0.25 (indicated by the dashed lines). Each point indicates the mean power and FDR across 20 replications and the error bars span one standard deviation either direction. In the top row, the response depends linearly on the features, and the feature statistics are calculated using the HRT with the LASSO. In the bottom row, the response depends non-linearly on the features, and the feature statistics are calculated using the HRT with random forest regression.

First, we look at how each procedure models the covariate distribution in Figure 1. In order to be valid knockoffs, the distribution of two knockoff features needs to be equal to that of the covariates. In this challenging example, each of the empirical knockoff methods fails to match the ground truth. In particular, DDLK and DeepKnockoffs are over-dispersed, while KnockoffGAN suffers from mode collapse. These findings for DeepKnockoffs and KnockoffGAN are similar to those reported by Sudarshan et al. (2020). Other than MASS, which directly fits a mixture of Gaussians, FLOWSELECT is the only method that matches the basic structure of the ground truth.

Figure 2 shows that the empirical knockoff procedures fail to control the FDR for both linear and nonlinear responses. One explanation for this lack of FDR control is the inability of the deep-learning-based methods to accurately model a knockoff distribution (c.f., Figure 1). As a result, the assumptions for the knockoff procedure will not hold, and FDR control is not guaranteed.

The effects of misspecification are clearly visible in the case of RANK, which approximates the mixture-of-gaussians data with a multivariate Gaussian. However, even MASS, when given access to the correct data distribution, does not achieve across-the-board FDR control. This highlights the potential sensitivity of knockoffs to parameter misfit even when the underlying distributional family of the features is known. This is confirmed by the fact that, when provided with the true parameters, the oracle Model-X maintains FDR control, though with significantly less power than FLOWSELECT. (c.f. Appendix H).

## 5.2 Semi-synthetic experiment with scRNA-seq data

In this experiment, we use single-cell RNA sequencing (scRNA-seq) data from 10x Genomics (10x Genomics, 2017). Each variable $X_{n,g}$ is the observed gene expression of gene $g$ in cell $n$. These data provide an experimental setting that is both realistic and, because gene expressions are often highly correlated, challenging. More background information about scRNA-seq data can be found in Agarwal et al. (2020).

We normalize the gene expression measurements to lie in $[0, 1]$, and we add a small amount of Gaussian noise so that the data is not zero-inflated. As in the semi-synthetic experiment from Sudarshan et al. (2020), we pick the 100 most correlated genes to provide a challenging, yet realistic example. We simulate responses that are both linear and nonlinear in the features. Figure 2 shows that FLOWSELECT maintains FDR control across multiple FDR target levels, feature statistics, and generated responses. In cases in which the knockoff methods control FDR successfully, FLOWSELECT has higher power in discovering the features the response depends on.

An advantage of knockoffs over CRT-based methods like FLOWSELECT is that the predictive model only needs to be evaluated once. Hence, while FLOWSELECT has a faster runtime than DDLK for this experiment, it is slower than DeepKnockoff and KnockoffGAN. However, Figure 2 shows that these two models fail to reliably control FDR and have much less power than FLOWSELECT; it is not clear how additional computational resources could be leveraged to improve the performance of these competing methods. A full table of runtimes on the scRNA-seq dataset can be found in Appendix F.

The need to compute a different predictive model for each feature within the CRT is mitigated by using efficient feature statistics such as the HRT (Tansey et al., 2021) and the distilled CRT (Liu et al., 2020). These methods fit a larger predictive model once, then evaluate either the residuals or test mean-squared-error for each feature individually. Moreover, the ability to scale to large feature dimensions $D$ is more limited by fitting the feature distribution than computational burden, a trait shared by both knockoff- and CRT-based methods.

FLOWSELECT provides asymptotic guarantees of FDR control assuming sufficient MCMC samples have been drawn for the p-values to converge. In this experiment, the consequence of terminating MCMC sampling before convergence is low power, rather than loss of FDR control (see Figure 7 in Appendix J). Even for small numbers of MCMC samples, the FDR stabilizes below the target rate, while the power steadily increases with the number of samples. Because the MCMC run is initialized at the true features, we speculate that the sampled features will be highly correlated with the true features in the beginning of the run, making it harder to reject the null hypothesis that a feature is unimportant.

## 5.3 Ablation Study

FLOWSELECT differs from the competing knockoff-based approaches in two ways: using normalizing flows with MCMC to model the feature distribution for sampling null features and using the CRT for feature selection. To illustrate the impact of each of these components separately, we compare to the procedure used in Tansey et al. (2021), which uses mixture density networks (MDNs) to model the complete conditional distribution of each feature $\mathcal{P}(X_j | X_{-j})$ separately. They then sample null features from these learned distributions directly and use the HRT for feature selection. Since both FLOWSELECT and this procedure utilize the HRT, this allows us to evaluate whether the performance improvement of FLOWSELECT over empirical knockoffs is solely due to use of the HRT.

We compare the MDN-based approach to FLOWSELECT on the mixture-of-Gaussians (Section 5.1) and scRNA-seq (Section 5.2) datasets. A plot of this comparison can be found in Appendix G. While the MDN-based approach was able to match the performance of FLOWSELECT on the scRNA-seq dataset, it failed to control FDR at any level on the Mixture-of-Gaussians dataset, indicating that MDNs are less flexible than normalizing flows. In aggregate, these results show that both the normalizing flows paired with MCMC and the use of the HRT for significance testing are key to the performance of FLOWSELECT.

## 5.4 Real data experiment: soybean GWAS

Genome-wide association studies are a way for scientists to identify genetic variants (single-nucleotide polymorphisms, or SNPs) that are associated with a particular trait (phenotype). We tested FLOWSELECT on a dataset from the SoyNAM project (Song et al., 2017), which is used to conduct GWAS for soybeans. Each feature $X_j$ takes on one of four discrete values, indicating whether a particular SNP is homozygous in the non-reference allele, heterozygous, homozygous in the reference allele, or missing. A number of traits are included in the SoyNAM data; we considered oil content (percentage in the seed) as the phenotype of interest in our analysis. There are 5,128 samples and 4,236 SNPs in total.

To estimate the joint density of the genotypes, we used a discrete flow (Tran et al., 2019). Modeling of genomic data is typically done with a hidden Markov model (Xavier et al., 2016); however, such a model may fail to account for long range dependence between SNPs, which a normalizing flow is better suited to handle. Having a more flexible model of the genome enables FLOWSELECT to provide better FDR control for assessing genotype/phenotype relationships. For the predictive model, we used a feed-forward neural network with three hidden layers. Additional details of training and architecture are presented in Appendix E.

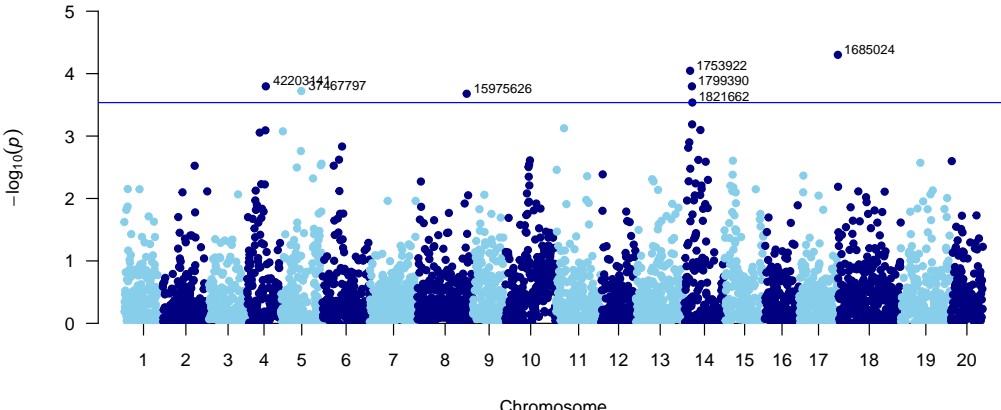

Figure 3: Manhattan plot for oil content in soybean GWAS experiment (Turner, 2018). $p$ is the estimated p-value from the FLOWSELECT procedure, and the blue line indicates the rejection threshold for a nominal FDR of $20\%$.

A graphical representation of our results is shown as a Manhattan plot in Figure 3, which plots the negative logarithm of the estimated p-values for each SNP. At a nominal FDR of 20%, we identified seven SNPs that are associated with oil content in soybeans. We cross-referenced our discoveries with other publications to identify SNPs that have been previously shown to be associated with oil content in soybeans. For example, FLOWSELECT identifies one SNP on the 18th chromosome, Gm18_1685024, which is also selected in Liu et al. (2019). FLOWSELECT also selects a SNP on the 5th chromosome, Gm05_37467797, which is near two SNPs (Gm05_38473956 and Gm05_38506373) identified in Cao et al. (2017) but which are not in the SoyNAM dataset. Sonah et al. (2014) identifies eight SNPs near the start of the 14th chromosome, and we select multiple SNPs in a nearby region on the 14th chromosome (seen in the peak of dots on chromosome 14 in Figure 3). However, the dataset in Sonah et al. (2014) is much larger ($\approx 47,000$ SNPs), which prevents an exact comparison. A list of all SNPs selected by our method is provided in Appendix E. For this experiment, FLOWSELECT tests over 4000 features in 10 hours using a single GPU. None of the empirical knockoff procedures (Sudarshan et al., 2020; Jordon et al., 2019; Romano et al., 2020) tested more than 387 features. This shows the potential for FLOWSELECT for high-dimensional feature selection with FDR control in a reasonable amount of time. Additional details about this experiment are available in Appendix E.

## 6  Discussion

FLOWSELECT enables scientists and other practitioners to discover features a response depends on while controlling false discovery rate, using an arbitrary predictive model; even large-scale nonlinear machine learning models can be utilized. By making fewer false discoveries for a fixed sensitivity level, FLOWSELECT can reduce the cost of follow-up experiments by limiting the number of irrelevant features considered. In contrast to the original model-X knockoffs method, FLOWSELECT does not require the feature distribution to be known a priori, nor does it require the feature distribution to have a particular form (e.g., Gaussian). Neither of these conditions are often satisfied in practice.

One limitation shared by both the conditional randomization test (CRT) and knockoffs is low power in cases in which important features are highly correlated with other important features. To mitigate this limitation, the CRT can be applied to test the significance of groups of correlated features rather than individual features. Within the FLOWSELECT framework, this entails modifying the MCMC step to draw null samples of groups of features conditioned on the others. The group's p-value can then be calculated with the same holdout randomization test (HRT) statistic used for testing individual

features. Group feature selection has also been explored for knockoffs (Dai & Barber, 2016; Liu et al., 2020).

Another limitation of FLOWSELECT stems from its reliance on normalizing flows. The flexibility of normalizing flows, though often beneficial, comes at a cost: sufficient training examples are needed to learn the feature distribution, limiting applicability in data-starved regimes. Fortunately, as we show in Appendix K, FLOWSELECT fares no worse than competing methods in low-data settings. In these regimes, FLOWSELECT could also use other density estimation techniques such as autoregressive models.

Furthermore, learning the feature distribution (potentially from limited data) is not the sole difficulty that the deep-learning-based knockoff methods face. To demonstrate that there are additional sources of difficult for knockoff-based methods, we gave DDLK, which typically fits the data distribution as part of its training procedure, access to the *the exact joint density*; neither the empirical FDR nor the power improved significantly (c.f. Appendix I). This result points to a failure of DDLK to enforce the swap property, which is a challenging task as the number of swaps grows exponentially with the number of features. FLOWSELECT, on the other hand, achieves FDR control under a different set of conditions that often are simpler to satisfy adequately in practice.

## Acknowledgments and Disclosure of Funding

Derek Hansen acknowledges support from the National Science Foundation Graduate Research Fellowship Program under grant no. 1256260. Any opinions, findings, and conclusions or recommendations expressed in this material are those of the author(s) and do not necessarily reflect the views of the National Science Foundation.

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
