# OpenReview forum: "Normalizing Flows for Knockoff-free Controlled Feature Selection"
_NeurIPS.cc/2022/Conference — NeurIPS 2022 Accept_

### Official Review · Reviewer_rjGq · 2022-07-08

**Rating:** 7
**Confidence:** 3
**Soundness:** 3 good
**Presentation:** 3 good
**Contribution:** 3 good

**Summary:**

The authors present a method, called FlowSelect based on normalizing flows to select features in a controlled fashion, meaning that the false discovery rate (FDR) is limited. They select features by first approximating the data distribution via the normalizing flow, and then compute a p-value for each feature explaining the data with respect to some model, e.g. a Lasso regression model or a Random Forest. Estimating the p-values involves sampling from the marginal distribution of the flow, which is done via MCMC.

To make sure that the FDR can be controlled, the authors prove that their estimates of the p-values are asymptotically correct, i.e. that the estimates converge to the true value almost surely.

In their experiments, they apply their method to a synthetic dataset, i.e. correlated Gaussians, a semi-synthetic dataset based on gene data, and a real-world dataset where they try to discover genes relevant for the oil content in soy beans. Their method outperforms several baseline procedures based on knockoffs, etc., and they are able to find SNPs relevant for the oil content which have also been confirmed by other studies.

**Questions:**

* What is the computational cost of computing the marginals of the flow for high dimensional systems?
* How do you choose the proposal for the MCMC procedure sampling from the marginals?
* Could you apply the other baseline methods to the genomic soy bean dataset?

**Limitations:**

It is not clear how well FlowSelect can be scaled up to very high dimensional datasets such as genomic data with 10k+ features.

**Strengths And Weaknesses:**

## Strengths

In general, the paper is well-structured and easy to read. I appreciate the background section, especially, because it guides the reader through the four lines of research the authors take ideas from and combine them to a new method.

FlowSelect combines several known and established methods in a creative way. It is well grounded from a theoretical perspective as the authors prove that their estimator for the p-values converges to the true value almost surely.

FlowSelect was put to test on three different datasets. It should to have high power on the synthetic and semi-synthetic dataset, higher or on par with all of the baselines, while having FDR as high or lower than the threshold, which is not the case for all the baselines. That the authors are also able to identify SNPs, which are relevant for the oil content in soy beans, is impressive.

## Weaknesses

Although the authors compare the SNPs they identify in the soy bean experiment to what has been found in other studies, they do not apply the baselines they used in the previous experiments to the same dataset, which makes their findings slightly less convincing.

Moreover, they give little insight in the computational cost of their method compared to competing procedures. Computing the marginals from the flow is only possible through MCMC, which can be very inefficient if the proposal distribution does not represent the actual distribution well. Especially when dealing with genomic data with 10k+ features this could render the method very inefficient. Unfortunately, the authors give very little insight into how they chose these proposal distributions and ensured that they represent the true marginals well.

## Conclusion

I'm in favor of accepting the article, although I wish that the others could clarify the concerns I raised above.

---

> ### Author Response · Authors · 2022-07-31
> **Author response to reviewer rjGq**
>
> We appreciate your positive review of our paper and feedback. We've included our response to specific points of yout review below:
>
> > Although the authors compare the SNPs they identify in the soybean experiment to what has been found in other studies, they do not apply the baselines they used in the previous experiments to the same dataset, which makes their findings slightly less convincing.
>
> Unfortunately, there is no way to directly evaluate the power/FDR on real datasets, as the true relevant features are essentially always unknown. Our third experiment is meant to show a relevant application of
> FlowSelect. While ground truth is not available like in our earlier experiments, we can compare our discoveries to those in previous studies. This is a common validation strategy in other controlled feature selection papers where ground truth is not known.
>
> > Moreover they give little insight in the computational cost of their method compared to competing procedures
>
> We show the comparative runtimes for each competing method in Appendix F for the scRNA-seq experiment. We comment on the runtime for the larger soybean dataset in Appendix E. FlowSelect finished in a reasonable amount of time in each of these cases.
>
> > Computing the marginals from the flow is only possible through MCMC, which can be very inefficient if the proposal distribution does not represent the actual distribution well. Especially when dealing with genomic data with 10k+ features this could render the method very inefficient. Unfortunately, the authors give very little insight into how they chose these proposal distributions and ensured that they represent the true marginals well.
>
> > How do you choose the proposal for the MCMC proposal sampling from the marginals?
>
> We describe the MCMC procedure in detail in Appendix D.1. We found that a simple Metropolis Hastings with a random walk Gaussian proposal worked well. This is because each MCMC chain targets a one-dimensional distribution, so we avoid the curse of dimensionality.
> It is true that each feature being tested requires its own MCMC chain, but these chains can be run in parallel. The convergence of the MCMC algorithm is explored in Appendix J.
>
> > What is the computational cost for computing the marginals of the flow for high-dimensional systems?
>
> The density evaluation cost for a normalizing flow is linear in the dimension of the variable, though for many flows such as the masked autoregressive flow (MAF) this is fully parallelizable on a GPU. FlowSelect also scales linearly with the number of features being tested, though this quantity is also fully parallelizable.
>
> > Could you apply the other baseline methods to the genomic soybean dataset?
>
> It is challenging to objectively compare the accuracy of methods without ground truth. The synthetic and semi-synthetic experiments are intended to offer a more objective point of comparison, while the soybean experiment is meant to showcase FlowSelect’s applicability to real problems. This follows the pattern in previous controlled feature selection papers.

---

> > ### Comment · Reviewer_rjGq · 2022-08-04
> > **Reply to the authors**
> >
> > I thank the authors for their insightful response. Since the most of the other reviewers scored the work similarly as I did, I will not change my score and vote for accepting this article.

---

### Official Review · Reviewer_ULQR · 2022-07-11

**Rating:** 4
**Confidence:** 4
**Soundness:** 3 good
**Presentation:** 2 fair
**Contribution:** 2 fair

**Summary:**

The paper considers the problem of multiple hypothesis testing with the FDR control. The work is situated in the model-X setting and is motivated by the limitations of the existing model-X multiple testing problems, with an emphasis on the knockoff-based methods. The paper proposes using the normalizing flow technique to estimate the conditional distribution of features, i.e., $X_j \mid X_{-j}$, which combine with CRT/HRT yields (approximately) correct p-values for conditional hypothesis testing. The p-values are subsequently fed to the BHq procedure, and a selection set is provided. The proposed method is evaluated with synthetic and semi-synthetic data, showing satisfying empirical results when compared to other methods that rely on inexact knockoff constructions.

**Questions:**

My major questions have been listed in the "strength and weaknesses" part. Some minor questions follow.

1. Line 55: should it be "In situations..."?

2. Line 157: is $\hat{\alpha}_j$ used as the p-value for both hypothesis $j$ and the ordered p-value?

3. Figure 1: what do the x and y-axis represent?

4. Line 212: what is $\Sigma_j$; line 213: what does the index $j$ mean in the definition of $\rho$.

5. Figure 2: it might be helpful to enlarge the fonts.

6. Figure 4: what do the column panels represent?



**Limitations:**

My comments on the limitations have been included in the "strengths and weaknesses" part. The authors have adequately addressed the potential negative societal impact of their work.

**Strengths And Weaknesses:**

Strength:
1. The paper finds a nice connection between a state-of-art  conditional density estimation method and the model-X feature
selection method.

2. The proposed method performs well empirically.

Weaknesses:
1. I find the motivation and comparison of the proposed method unsuitable. The paper tries to see the proposed method as an alternative method for the knockoff-based method; but at its best (i.e., when the mapping G is exactly known), the proposed method does not guarantee the FDR control due to the dependence between the p-values. Further, the target distribution knockoffs (or inexact knockoff machines) tries to learn is different from what the proposed method tries to learn: the former is a knockoff distribution satisfying the ``swapping invariant'' property while the latter is simply the conditional distribution. I would suggest framing the proposed method as a way of implementing CRT/HRT when the feature distribution is not exactly known.

2. Following the previous comment, I find the comparison in the simulations a bit unfair: methods w/ different guarantees;
one-bit p-values versus multi-bit p-values. For example, it might be more appropriate to consider using the existing deep method to learn $X_j \mid X_{-j}$ and apply CRT/HRT.

3. The presentation of this paper can be improved.

a) the notation is not consistent throughout the paper. For example, the dimension is represented as d in line 44, then D in line 45, and then p in line 103.

b) The description of the existing method is a bit hard to follow. For example, it may help to briefly explain where the test data and training data come from; Theorem 1 involves the index n, and it would help to explain what is this n in the actual flow used (is it M?).

---

> ### Author Response · Authors · 2022-07-31
> **Author response to reviewer ULQR**
>
> We appreciate the detailed feedback in your review. We’ve included a point-by-point response below.
>
> > I find the motivation and comparison of the proposed method unsuitable. The paper tries to see the proposed method as an alternative method for the knockoff-based method; but at its best (i.e., when the mapping G is exactly known), the proposed method does not guarantee the FDR control due to the dependence between the p-values. Further, the target distribution knockoffs (or inexact knockoff machines) tries to learn is different from what the proposed method tries to learn: the former is a knockoff distribution satisfying the ``swapping invariant'' property while the latter is simply the conditional distribution. I would suggest framing the proposed method as a way of implementing CRT/HRT when the feature distribution is not exactly known.
>
> The best-case theoretical guarantees for knockoffs are superior to those of the CRT **if the data distribution is known, and if the knockoff distribution can be sampled exactly**. These two conditions are often not met in practice. Our submission is concerned with the setting in which these conditions are not met. In this challenging setting, no methods -- neither knockoff-based methods nor FlowSelect -- guarantee FDR control without additional assumptions. Knockoff-based methods make one set of assumptions (e.g., the knockoff distribution can be fitted and sampled “well enough”) whereas our method makes a different set of assumptions (e.g., that asymptotics are relevant to the analysis of finite datasets; that the conditions of Benjamini-Hochberg are satisfied well enough).
>
> A key finding of our paper is that our method (based on the CRT) outperforms the deep-learning knockoff methods empirically. This indicates that the conditions for a valid, swap-invariant knockoff distribution are not being satisfied well enough in practice.
>
> We agree that FlowSelect is a way of implementing the Conditional Randomization Test (CRT) in the case where the feature distribution is not exactly known. The CRT has FDR control under the assumptions of the Benjamini-Hochberg (B-H) multiple testing procedure. B-H doesn’t require that the p-values are independent; it just requires that they are positively correlated.
>
>
> > Following the previous comment, I find the comparison in the simulations a bit unfair: methods w/ different guarantees; one-bit p-values versus multi-bit p-values.
>
> Although the CRT and Knockoffs are different methods, they both share the same goal in selecting features while maintaining false discovery rate (FDR) control. Thus, we believe it is fair to compare how well each method does at this task empirically. In fact, our experimental setup was directly based on those from previous knockoff papers, in particular Sudarshan et al (2020).
>
> > It might be more appropriate to consider using the existing deep method to learn $X_j | X_{-j}$.
>
> We show a comparison to the holdout randomization test (HRT) from Tansey et al (2021) in Appendix G, which uses mixture-density networks to directly learn $X_j | X_{-j}$.
>
> > Theorem 1 involves the index n, and it would help to explain what is this n in the actual flow used (is it M?).
>
> The index “n” signifies a sequence of normalizing flows that converge to the true mapping. It encodes the assumption that one can learn a mapping that is arbitrarily close to the true mapping.
>
> > The presentation of this paper can be improved...
>
> We appreciate spotting the notation inconsistencies and typos, and we've made the following fixes in the rebuttal revision:
>
> - We corrected the inconsistency in the feature dimension.
> - We removed the line describing the sorted p-values ($\alpha_j$), as the selection threshold definition does not require them to be sorted.
> - We described the splits for the experiments (90% of data is used for training, 10% for evaluating the feature statistic).
> - In Figure 1, we’ve added a clarification that the x-axis is the first component and the y-axis is the second component.
> - In Figure 2, we've enlarged the fonts.
> - In Figure 4, we have added more clarification what the columns are.

---

> > ### Comment · Reviewer_ULQR · 2022-08-08
> > **Response**
> >
> > I would like to thank the authors for the response. I agree that in the scenario considered in this paper, the method of knockoffs does not offer theoretical FDR control, and the proposed method performs better empirically. However, the proposed method does not have a theory for the FDR control either---to the best of my knowledge, there is no result showing that the CRT p-values are
> > PRDS (also valid p-values are only guaranteed asymptotically given the mapping $G(\cdot)$). That said, the empirical advantage of the proposed method is promising---it is just that I find the theoretical guarantee insufficient.

---

> > > ### Author Response · Authors · 2022-08-09
> > > **Reply to reviewer ULQR**
> > >
> > > We thank the reviewer for continuing this discussion, and for the positive comments about the strength of our experimental results. It appears that we were able to address your original concern about the suitability/fairness of our comparisons.
> > >
> > > We see Theorem 1 as sufficient to advance our narrative. The hypothesis motivating our work is that empirical knockoff-based approaches do not adequately satisfy the swap condition in practice, and that a method with different (but not strictly fewer) conditions, which are potentially less onerous to enforce, may perform better. Theorem 1 helps to explain the conditions under which our proposed method achieves FDR control.
> > >
> > > We agree that Theorem 1 does not show that the proposed method has strictly fewer conditions than empirical knockoff-based approaches—but having strictly fewer conditions doesn’t seem necessary to advance our narrative. Theorem 1 establishes that the proposed method requires different conditions. It’s only through empirical work that we demonstrate that these alternative conditions can often be enforced well enough, whereas the conditions of knockoff-based competitors cannot.
> > >
> > > We will incorporate this discussion into the manuscript to give greater context for the role of Theorem 1.

---

### Official Review · Reviewer_qGxr · 2022-07-11

**Rating:** 6
**Confidence:** 2
**Soundness:** 3 good
**Presentation:** 3 good
**Contribution:** 3 good

**Summary:**

This paper proposes a new variable selection method called FLOWSELECT to perform controlled feature selection that does not suffer the problems existed in Knockoffs-based methods. Asymptotically,  the proposed method computes valid p-values. Empirically, FLOWSELECT consistently controls the FDR on both synthetic and semi-synthetic benchmarks, whereas competing knockoff-based approaches do not.

**Questions:**

[Fan Y. et al] propose a modified model-X knockoffs method, called graphical nonlinear knockoffs (RANK), to accommodate the unknown covariate distribution. The related theoretical results verify the robustness of the RANK by showing that the false discovery rate (FDR) is asymptotically controlled at the target level and the power is asymptotically one with the estimated covariate distribution.

+ Does the RANK overcome the first problem often occurred in the knockoff-based method emphasized by the authors?

+ It would be appreciated if the author could discuss the difference between this method and the above method.

+  Can rank method be added as a baseline in the numerical experiments？
[Fan Y. et al] Large-Scale Inference With Graphical Nonlinear Knockoffs, JASA.

**Limitations:**

All my concerns are presented in “Weakness” and “Questions”. Moreover, I do not have any concern on negative social impact.

**Strengths And Weaknesses:**

Strength:
+  The paper is well-motivated and well-written.
+  The proposed method has nice theoretical guarantees and sufficient empirical evaluations.


Weakness:
+ In fact, some work on knockoff has improved the two problems highlighted by the author that lead to the failure of traditional knockoffs-based methods, e.g., “Fan Y. et al. RANK: Large-Scale Inference With Graphical Nonlinear Knockoffs.”. However, the authors does not discuss this related literature.

---

> ### Author Response · Authors · 2022-07-31
> **Author response to reviewer qGxr**
>
> We appreciate the reference to the RANK method for knockoffs. RANK assumes the features are generated from a multivariate Gaussian with an unknown, sparse precision matrix. In contrast, the deep-learning-based approaches and FlowSelect do not assume a specific parametric form for the feature distribution. Our comparison of the proposed method to “Knockoffs for the MASS”, which models predictors as a mixture-of-Gaussians, suggests that RANK is unlikely to perform well outside of the multivariate Gaussian setting.
>
> To further investigate, we’ve implemented RANK and obtained results for both our mixture-of-Gaussians experiment and our scRNA-seq experiment. On the mixture-of-Gaussians dataset, we found that RANK did not control the false discovery rate (FDR), like the other knockoff-based approaches. RANK performs a bit better in the scRNA-seq experiment, but still does not maintain consistent FDR control. This lack of performance is likely due to the multivariate Gaussian assumption being violated. We’ve updated Figure 2 in the rebuttal revision to show these new results.

---

> > ### Comment · Reviewer_qGxr · 2022-08-08
> > **Thanks for your response!**
> >
> > I thank your detailed response for addressing my concerns. I have also read other reviewers' comments carefully.  I will keep my score as is and I'd love to hear from Reviewer  ULQR  about his thoughts.

---

### Official Review · Reviewer_vcjW · 2022-07-15

**Rating:** 7
**Confidence:** 3
**Soundness:** 4 excellent
**Presentation:** 4 excellent
**Contribution:** 3 good

**Summary:**

This work considers the controlled feature selection problem, or selecting a small number of important features subject to FDR control. The proposed method, FlowSelect, builds on the existing literature of conditional randomization tests and holdout randomization tests, as well as normalizing flows from the deep generative modeling literature.

In simple terms, FlowSelect uses normalizing flows to learn a data distribution and MCMC to sample individual held-out features from their conditional distribution; it then passes these samples through a single model fit on the original data to generate samples from the null distribution, and it then compares these to the observed test statistic calculated using the observed data (with the rejection threshold set according to Benjamini-Hochberg).

The proposed approach is justified using an asymptotic argument: normalizing flows are theoretically capable of modeling any distribution given enough transforms, and MCMC should be able to sample from the learned distribution given enough samples. The empirical results are quite promising, FlowSelect provides better results than several competing knockoff methods across two types of synthetic tasks, as well as reasonable results on a real scRNA-seq dataset.

**Questions:**

- Would the authors be able to include additional results showing how FlowSelect (and the baselines) perform given a smaller dataset? I would be curious to see if the normalizing flows approach scales better or worse than the other methods, particularly those that use deep learning
- Based on a quick Google search, it seems there are other papers that have used MCMC to sample from distributions parameterized by normalizing flows (e.g., https://arxiv.org/abs/2105.12603). Are the authors aware of these, and would it make sense to comment on the similarities/differences between this and some existing use cases of this trick?
- As mentioned above, the asymptotic argument for why FlowSelect should work is somewhat optimistic and doesn't quite explain why we should expect this approach to work in the finite regime (finite transformations in the flow model, finite MCMC sampling). Can the authors comment on whether they might be able to refine their analysis, or perhaps include a discussion of how to verify if either part is working "well enough" ? (E.g., metrics for whether the normalizing flow fit the distribution well, established techniques for deciding if MCMC has been run for long enough.)
- Could the authors add a note about why it's important to add Gaussian noise in the scRNA-seq experiment (what the issue is with zero-inflation)?
- This isn't a weakness in FlowSelect specifically, but perhaps in all CRT-based methods that may be helpful to address: it seems like a CRT may fail to identify an important feature that is correlated with another important feature, because removing either from the model should have no impact on the test statistic. Could the authors explain this and why it is/isn't a problem, and if necessary add a note (perhaps in section 2)?

**Limitations:**

A couple of the questions above get at limitations that might be helpful to discuss:
- Ability to identify important features that are strongly correlated with one another (in the worst case, identical)
- Limited applicability of theory to practical usage
- Unclear scaling with dataset size


**Strengths And Weaknesses:**

### Strengths

- The method combines known methods for feature selection under FDR control with two new techniques (at least in this subfield): normalizing flows and MCMC to sample from the learned distribution
- The asymptotic argument makes clear why this approach should work, at least in scenarios with a large amount of data, a large flow model, and sufficient MCMC samples
- The empirical results are quite promising
- The ablation study showed that multiple methodological innovations are important for helping FlowSelect outperform the baselines (the normalizing flows + MCMC approach for handling held-out features, as well as the HRT)

### Weaknesses

- The theory is somewhat optimistic: the p-values converge only under two asymptotic results. It's natural to develop asymptotic theory first as this is often easier, but I wonder if there's any way the theory could be refined to more closely reflect how FlowSelect performs in practice, or at least help control whether either of the key steps is working well enough (more on this in the questions section below)


Nits:
- This may not be a mistake, but I wasn't sure of the reason for the +1 in the summation and denominator of eq. 5
- There's a typo on line 252, "to have support in..."
- I may have missed this, but what exactly was the test statistic used in the experiments? The caption of figure 2 says the statistics were calculated using either Lasso or random forest; was it the model's predictive accuracy?

---

> ### Author Response · Authors · 2022-07-31
> **Author response to Reviewer vcjW**
>
> We appreciate your positive review, and we have included a response to each item of feedback below.
>
> > The theory is somewhat optimistic: the p-values converge only under two asymptotic results...
>
> > Can the authors comment on whether they might be able to refine their analysis, or perhaps include a discussion of how to verify if either part is working well-enough?
>
> Like much statistical theory, Theorem 1 is an asymptotic result. Fortunately, each of the parts on which it relies - the quality of the normalizing flow and the convergence of MCMC - can be checked.
>
> First, one can check whether samples from the normalizing flow resemble those of the true distribution, which we show for the mixture-of-Gaussians experiment in Figure 1. One can also check whether the trained flows map the features to a standard Gaussian. We’ve added an appendix (Appendix L) to the rebuttal revision showing the distribution of these mapped features in “flow space”. The mapped features from the scRNA-seq dataset are visually indistinguishable from true samples from a standard Gaussian. The mapped features from the mixture-of-Gaussian are close to standard Gaussian, but appear to have a slight multi-modal structure. The mapping could possibly be improved with specific flow architectures for multi-modal distributions. However, the strong empirical power and FDR results for the mixture-of-Gaussians dataset suggest that FlowSelect is robust to slight deviations from the true mapping.
>
> Second, the number of MCMC samples is directly under control of the user. Convergence of the p-values can be checked visually, or through a MCMC diagnostic such as Gelman-Rubin. We demonstrate in Appendix J the effect of not having a sufficient number of MCMC samples. The practical effect is that the empirical number of false discoveries remains below the threshold, but the power suffers.
>
> > Clarify test statistic used in experiments
>
> We use the holdout randomization test (HRT) with the predictive log-likelhood from the LASSO (for the linear response) and the predictive negative mean-squared error from the random forest (RF) for the non-linear response. We’ve added this description to the main text in the experiments section in the rebuttal revision.
>
> > Would the authors be able to include additional results showing how FlowSelect (and the baselines) perform given a smaller dataset?
>
> We show in Appendix K a comparison of all the methods in the mixture-of-Gaussians dataset, but with fewer observations (n=2000) and less correlation among the features. We found that all methods did well on the linear response (similar power and FDR control), but no method held FDR control with the non-linear response. The good performance in the linear setting can be explained by the LASSO feature statistic shrinking most null feature importances to zero. However, since FDR control should hold regardless of the response, these findings suggest that none of the methods do well in this relatively low-data regime. This exposes the trade-off of using flexible, black-box models that do not exploit specific structure in the feature distribution - more data is needed to learn this structure.
>
> > ...there are other papers that use MCMC to sample from distributions parameterized by normalizing flows
>
> The linked paper in question is tackling a fundamentally different problem. They are focused on the general problem of MCMC sampling from an intractable density that can be evaluated up to a normalizing constant. They use normalizing flows to improve the MCMC kernel for better performance at this task.
> In contrast, we use normalizing flows to estimate the density of the joint feature distribution from samples. Then, we utilize MCMC to sample the conditional distribution of each feature.
>
> > ... why it's important to add Gaussian noise in the scRNA-seq experiment ...
>
> Raw scRNA-seq data is count data, but it is common in practice to normalize and rescale these counts so that they are approximately Gaussian. In addition to rescaling these counts, we add a bit of noise to them to ensure they are continuous.
>
> > The CRT may fail to identify an important feature that is correlated with another important feature
>
> We agree that features that are erroneously marked as irrelevant are likely those correlated with other important features.
> In this setting, one can modify the CRT to test the joint importance of pairs of features. Using FlowSelect, this would simply entail changing the MCMC step to sample pairs of features conditioned on the rest, and then using the hypothesis randomization test (HRT) to calculate a joint importance statistic. The resulting p-values could then be used to select relevant pairs of features in the same manner as before. We are at the page limit for the rebuttal revision, but we will add a paragraph explaining this to the discussion (section 5) in the camera-ready version.

---

> > ### Comment · Reviewer_vcjW · 2022-08-07
> > **Thanks for author response**
> >
> > Thanks to the authors for their detailed responses. My opinion of the paper remains positive so I'll keep my score as is. A couple quick thoughts on what the authors wrote in their response:
> >
> > **Normalizing flow quality and MCMC convergence.** This is helpful, thanks for the clarification. A brief discussion of these points in the paper seems important for practical usage of FlowSelect by other researchers.
> >
> > **Possible failure to identify correlated features.** Thanks for being willing to add this discussion. I'm not sure testing pairs of features solves the problem, as you could have a group of $m$ correlated features for $m > 2$, in which case you may need to generalize FlowSelect to deal with feature subsets of size $m$. Any discussion of this issue/limitation would be helpful to discuss, although it's clearly a challenging problem and not unique to this approach.

---

### Meta-Review · Area_Chair_1u7W · 2022-08-27

**Recommendation:** Accept
**Confidence:** Certain

**Metareview:**

This paper describes how to use normalizing flows for selecting features in a way that controls the type-1 error by using a normalizing flow along with MCMC to sample from the null distribution. The majority of the reviewers were positive, however the most confident reviewer was negative. From taking a look at that reviewers concerns, I tend to agree with most of them.

The paper is titled knockoff-free, which means in the context of this paper that both 1) 1-bit p-values are not used and 2) The full knockoff property is not required, only sampling from complete conditionals are required. Most of the experiments compare knockoff methods to the proposed approach, so it's not clear if 1) 1-bit p-values are not great or 2) the model-X process/complete conditional sampling process is better with normalizing flows. The former point is known and the latter point on the best way to sample from the complete conditionals is really the value.

If we take the paper as,

1) complete conditionals are 1-D
2) MCMC can be used to sample from a 1-D unnormalized density
3) Simple MCMC won't be bad because the problem is 1-D

-> Any likelihood based deep generative model can be used to sample complete conditionals

then it's a solid paper.

On the other hand, the belief that flows are the correct choice versus other likelihood-based deep generative models is harder to take as there's only a comparison with a mixture density network used in the original HRT paper. Also from other uses of these models, different models are better in different situations. I'd suggest a heavy discussion in the paper on this point at the minimum. Maybe even a reframing of the paper is needed.

Finally, for the test statistic, the HRT may not be the best choice for work like this paper that studies the problems with estimating X-distribution. The  paper "CONTRA: Contrarian statistics for controlled variable selection" at AISTATS 2021 shows that the HRT test statistic is more sensitive to model-X estimation errors than a simple mixture statistic that doesn't give up much power. The choice of test statistic also merits some discussion in step 3.

**Award:**

No

---

### Decision · Program_Chairs · 2022-09-14

Accept